# A Combined Persistence and Physical Approach for Ultra-Short-Term Photovoltaic Power Forecasting Using Distributed Sensors

**DOI:** 10.3390/s24092866

**Published:** 2024-04-30

**Authors:** Yakov Malinkovich, Moshe Sitbon, Simon Lineykin, Kfir Jack Dagan, Dmitry Baimel

**Affiliations:** 1Department of Electrical Engineering and Electronics, Ariel University, Ariel 40700, Israel; jakov.melinkov@msmail.ariel.ac.il (Y.M.);; 2Department of Mechanical Engineering and Mechatronics, Ariel University, Ariel 40700, Israel; 3Department of Electrical and Electronics Engineering, Shamoon College of Engineering, Beer-Sheva 84100, Israel; dmitrba@sce.ac.il

**Keywords:** photovoltaic (PV) fields, cloud cover, energy storage system, nowcasting

## Abstract

This paper presents a novel method for forecasting the impact of cloud cover on photovoltaic (PV) fields in the nowcasting term, utilizing PV panels as sensors in a combination of physical and persistence models and integrating energy storage system control. The proposed approach entails simulating a power network consisting of a 22 kV renewable energy source and energy storage, enabling the evaluation of network behavior in comparison to the national grid. To optimize computational efficiency, the authors develop an equivalent model of the PV + energy storage module, accurately simulating system behavior while accounting for weather conditions, particularly cloud cover. Moreover, the authors introduce a control system model capable of responding effectively to network dynamics and providing comprehensive control of the energy storage system using PID controllers. Precise power forecasting is essential for maintaining power continuity, managing overall power-system ramp rates, and ensuring grid stability. The adaptability of our method to integrate with solar fencing systems serves as a testament to its innovative nature and its potential to contribute significantly to the renewable energy field. The authors also assess various scenarios against the grid to determine their impact on grid stability. The research findings indicate that the integration of energy storage and the proposed forecasting method, which combines physical and persistence models, offers a promising solution for effectively managing grid stability.

## 1. Introduction

Renewable energy refers to energy sources that are inexhaustible or renewable, drawing power from processes that occur constantly and are predicted to continue indefinitely in the future. Examples include solar energy, wind energy, geothermal processes, and water movement. Unlike energy production based on fossil fuels (such as coal, natural gas, and oil) or nuclear fusion, renewable energy minimizes the substances used for energy production [1]. The versatility of renewable energy sources, particularly solar power, has led to innovative applications beyond traditional power generation, such as solar fencing for agricultural and perimeter security [2]. According to the International Energy Agency, the growth of renewables has accelerated in recent years due to the decreasing costs of wind energy and photovoltaic technology, as well as continued government support [3]. One of the main targets for renewable energy is the electricity sector, and it is predicted that renewable energy sources will meet most of the world’s energy requirements. There has been steady growth in the worldwide deployment of solar PV systems, with a 2021 capacity of approximately 107 GW [1].

The Earth’s surface receives approximately 1.5*1018 kWh/a of solar energy every year, which is ten thousand times the current power consumption needs of all nations [4]. The increasing global demand for clean and sustainable energy sources has led to the widespread adoption of renewable energy technologies. Among these, solar PV panels have emerged as a popular and environmentally friendly solution for generating electricity. PV systems harness the sun’s energy by converting sunlight into electrical power, providing a clean and inexhaustible energy resource. However, the intermittent nature of solar power, influenced by factors such as cloud cover, poses challenges to the stability and reliability of power grids [5].

PV panels are made up of PV cells that use natural, inexhaustible sunlight and convert it into electrical energy. Since each PV cell produces approximately 5 W and 0.5 V DC, cells placed in a series PV arrangement are an unreliable resource, so grid integration is not trivial [6].

Accurate power prediction for PV systems is crucial to overcoming these challenges and ensuring the seamless integration of solar energy into the grid. Power forecasting allows grid operators and utilities to anticipate fluctuations in solar power output and make informed decisions about energy management. In general, power prediction can be classified into different types based on the forecasting horizon, including nowcasting, ultra-short-term, short-term, medium-term, and long-term forecasts. Nowcasting and ultra-short-term forecasts are particularly important for managing the variability of renewable energy generation and maintaining grid stability. Despite the critical importance of accurate and efficient forecasting methods for PV power output, the existing literature, including recent reviews, predominantly focuses on models that require extensive computational resources or rely heavily on historical data, which may not always be available or accurate for all locations [7]. Various forecast models have been developed to predict solar power production under different weather conditions, including persistence models, physical models, and a combination of both. Persistence models rely on the assumption that recent observations of power output will persist in the near future, while physical models use numerical weather prediction data to estimate the impact of weather conditions on solar power generation. The method presented in this paper, utilizing distributed sensors for ultra-short-term photovoltaic power forecasting, is not found in the existing literature. By integrating a combined persistence and physical approach, this method offers a novel solution that addresses the gap in efficiently predicting the impact of cloud cover on photovoltaic fields in real time. The future time period for output forecasting or the time duration between the actual and effective time of prediction is the forecast horizon [8]. Some researchers prefer three categories of the forecast horizon: short-term, medium-term, and long-term [9]. Others have added a “fourth” category that is useful in designing PVs integrated with a better energy management system, unit commitment, power scheduling, and dispatching, which is named the “very short-term or ultra-short-term forecast horizon” [10,11]. Accurate forecasting is crucial for enhancing the efficiency and reliability of PV-BESS systems. Precise solar power predictions are essential for managing BESS charging and discharging cycles, which, in turn, improves energy efficiency and grid stability. Therefore, there is a clear need for advanced forecasting models to better integrate solar power with energy storage solutions [12]. This approach adds significant value by enhancing the accuracy of power predictions in the ultra-short term, enabling more effective integration of solar power into the grid and improving grid stability.

## 2. System Description

A photovoltaic field typically spans a large area, depending on the planned power production capacity. Such a field serves as a single unit of power supply connected to the electricity grid. As depicted in Figure 1, these installations are often arranged in rectangular or square configurations, optimizing the use of space for ease of implementation, maintenance, and control. The panels within the field are strategically interconnected using DC/DC converters, which may be connected to a single panel or multiple panels to regulate power and voltage effectively. The larger the power output of the field, the more significant its impact on the power grid. Slow power fluctuations within such a field arise due to changes in the sun’s angle, temperature variations, wind, dust, and other factors. However, rapid changes are most notably caused by cloud cover. In Figure 1, the thick blue line represents the placement of panel sensors in the field. This figure also shows how clouds passing over the field can temporarily obscure either entire sections or just parts of the panels, leading to immediate and significant decreases in power generation. These dynamics underscore the variable nature of solar energy collection. To address these fluctuations and enhance the system’s overall efficiency, there is increasing interest in incorporating energy storage systems. Such systems can improve the photovoltaic field’s responsiveness and allow for the sale of energy during peak demand hours, thereby stabilizing the supply to the grid. In the depicted PV field, solar cells are connected in series and parallel arrangements to meet specific voltage or current demands. DC/AC converters are then distributed across the field to facilitate a reliable connection to the electricity grid, ensuring a consistent and efficient power output from the solar array.

## 3. Problem Formulation

PV technology, as a source of power, can be subject to fluctuations and may decrease its output within seconds. As a result, grid integration and power quality are crucial factors, particularly as numerous advanced countries have made a conscious effort to transition toward renewable energy sources. Real-time factors, such as weather conditions, can significantly impact PV power generation, making the construction of a reliable prediction system a complex endeavor [13].

This paper will specifically focus on the effects of cloud cover on PV output. Accurate forecasting of solar power generation depends on various factors, including appropriate forecasting horizons, suitable forecast model inputs, and reliable performance estimation methods.

A better understanding of these factors and their interdependencies is essential for developing a more comprehensive and accurate prediction system for PV power generation. Such a system can play a pivotal role in managing grid stability and ensuring the effective integration of solar power into the overall energy mix. In turn, this will support ongoing global efforts to increase the adoption of renewable energy sources and reduce our reliance on fossil fuels [3].

*Forecasting horizons:* The forecast horizon refers to the future time period for output forecasting or the time duration between the actual and effective time of prediction [8].

### 3.1. Now Forecasting

“Solar nowcasting” aims to address the inconsistency in solar power generation by providing short-term forecasts of anticipated power generation capacity. This new subdomain of solar forecasting focuses on predicting intra-minute solar variability, which is largely affected by local clouds. To achieve this, solar nowcasting utilizes high-frequency sensors and predicts shadow or cloud movements in both time and space. These forecasts can be of great benefit to electricity marketing and pricing, power-smoothing processes, real-time electricity-dispatch monitoring, and PV storage control [6].

### 3.2. Very Short-Term Forecasting

Short-term forecasting is a valuable tool in the electricity market for making decisions related to economic load dispatch and power system operation. It is also crucial in the management of renewable-energy-integrated power systems. The typical time frame for short-term forecasting ranges from 30 to 360 min, although some studies consider periods of one to several hours, one day, or even up to seven days as falling within the short-term forecast horizon [14]. For example, it has been proposed that electric load patterns should be forecasted two days in advance to enable the effective scheduling of power plants and planning of transactions. Short-term forecasting also enhances the security of grid operations by enabling grid operators to anticipate changes in demand and take proactive measures to maintain stability and reliability. Enhanced ultra-short-term PV forecasting increasingly utilizes machine learning methods, including CNN models, to integrate meteorological data, aligning with our focus on predictive accuracy in immediate timeframes [7,15]. Furthermore, the integration of IoT sensor data into forecasting models presents an innovative approach to enhancing prediction accuracy. This method leverages on-site IoT sensors for detecting environmental changes specific to the solar panel location, offering a more refined analysis for very short-term forecasting needs [16]. Overall, short-term forecasting plays a vital role in ensuring the efficient and effective operation of power systems and grid infrastructure [17,18,19].

### 3.3. Medium-Term Forecasting

Medium-term forecasting spans 6–24 h and is crucial for efficient grid management and renewable energy integration [14]. Current state-of-the-art methods include numerical weather prediction (NWP) models for solar irradiance and meteorological variable predictions, machine learning techniques such as artificial neural networks and support vector machines for pattern recognition and data-driven forecasts, and hybrid approaches that combine the strengths of both NWP models and machine learning techniques. Advancements in forecasting models, including deep learning approaches and graph convolutional networks (GCNs), further refine prediction capabilities by capturing temporal and spatiotemporal patterns within the generated data [20]. For accurate medium-term PV power forecasts, utilizing high-quality data sources and continuously refining forecasting models is essential [8,21].

### 3.4. Long-Term Forecasting

Long-term forecasts anticipate scenarios more than 24 h ahead, with some sources even defining periods of a month to a year as long-term forecasts [14]. These prediction horizons are appropriate for long-term power generation, transmission, distribution, and solar energy allocation and account for seasonal trends [8]. Key methods include scenario-based forecasting to evaluate potential future outcomes, techno-economic modeling to analyze growth and adoption, and time-series analysis and econometric models for trend-based projections [22,23].

### 3.5. Dependence of PVPF Accuracy on the Time Horizon

The accuracy of photovoltaic power forecasting is highly dependent on the time horizon considered. While long-term and medium-term forecasts serve specific purposes, nowcasting, which focuses on very short-term forecasts (typically from a few minutes to a few hours), has emerged as a highly accurate approach to PV power prediction. This review discusses the advantages of nowcasting and its importance in optimizing PV power. The advantages of nowcasting include higher accuracy by using real-time data, enhanced grid stability and power management, and economic benefits [24]. Therefore, it is clear that error is minimized when the time horizon is minimal [8].

PV output performance is mainly affected by weather conditions such as solar irradiance (the amount of sunlight reaching the panels), temperature, cloud cover, aerosol distribution, and wind speed. When utilizing real-time sensor data for PV power predictions, weather classification is not necessary. This is because the real-time sensor data already reflect the current weather conditions and their impact on the system’s performance. With these data, PV output can be predicted directly without the need for additional weather classification steps. Moreover, real-time sensor data provide more accurate and up-to-date information about the environmental factors affecting the panels, making weather classification redundant [24].

### 3.6. Clouds

Cloud cover plays a crucial role in forecasting solar PV power generation because it directly affects the amount of solar radiation reaching the Earth’s surface. Accurate predictions of cloud movements and coverage can significantly improve the reliability of PV power forecasts, helping grid operators manage power distribution more effectively and enabling PV system owners to maximize their energy output and profits [25]. Most investigators have used satellite images for cloud analysis [26]. However, these images have limited accuracy in regard to analyzing regional or low cloud formations due to their low partial and temporal resolutions. As a result, they are not very useful in ultra-short-term forecasts [11]. The following factors play important roles in cloud cover in PV power forecasting: *Variable solar radiation*: Clouds can cause rapid fluctuations in solar radiation levels by blocking, reflecting, or scattering sunlight [27]. The presence and movement of clouds lead to variations in the intensity and distribution of solar radiation, making it challenging to predict PV power output accurately. *Spatial and temporal variability*: Cloud cover can vary significantly across different locations and time scales. Cloud formation, movement, and dissipation depend on various factors, such as atmospheric conditions, topography, and local weather patterns [17]. Accurate PV power forecasting must consider both the spatial and temporal aspects of cloud cover. *Different cloud types*: Clouds can be classified into various types based on their altitude, thickness, and optical properties. Each cloud type impacts solar radiation differently, with some causing more significant reductions in solar radiation than others [6]. Forecasting models need to consider the specific cloud types present in a given location to provide accurate PV power predictions [28]. *Sky imaging and cloud tracking*: High-resolution sky cameras can capture images of the sky and track cloud movements in real time. These images are then used in PV power forecasting models to account for the impact of cloud cover on solar radiation levels. *Numerical weather prediction*: By analyzing sky images, nowcasting models can quickly predict changes in solar radiation levels and PV power output in response to evolving cloud conditions over the next few minutes to hours [27]. *NWP models*: NWP models simulate atmospheric conditions and predict weather patterns, including cloud cover. These models can be used to forecast solar radiation levels and, in turn, PV power output. However, NWP models may have limitations in accurately predicting cloud cover on short time scales and at high spatial resolutions. *Machine learning and artificial intelligence*: Advanced machine learning algorithms and artificial intelligence techniques can analyze historical and real-time data to predict cloud cover, solar radiation levels, and PV power output. These methods can learn from past data to improve the accuracy of their predictions over time. *Ensemble forecasting*: Combining multiple forecasting models can help reduce uncertainty and improve the overall accuracy of PV power predictions. Ensemble forecasting accounts for the strengths and weaknesses of different models to provide a more robust forecast of solar radiation and PV power output [29].

### 3.7. Forecast Types

There are several types of forecasting models used for predicting PV power output, each with its strengths and weaknesses. Persistence models rely on the assumption that current conditions will persist in the near future. Their main advantage is that they are simple and computationally efficient. Physical models, such as numerical weather prediction (NWP) models, simulate atmospheric processes to predict solar radiation and cloud cover, providing a more detailed representation of weather patterns. However, these models can be limited by their spatial and temporal resolutions. Statistical models use historical data and mathematical techniques to predict PV power output based on correlations between input variables (e.g., weather data) and output variables (e.g., PV power generation). Machine learning models, a subset of statistical models, employ advanced algorithms to learn from large datasets and adapt their predictions over time, providing potentially more accurate forecasts as they learn from new data [30].

Persistence forecasts hinge on the extrapolation of prevailing conditions into future horizons. The persistence method is the simplest type of forecast and is the most common reference model for short time-horizon forecasts [29]. For solar irradiance prediction, the model assumes clear sky conditions and that the irradiance I at a given time *t* (delay0) will be the same as that at the previous time step *t* − 1 (delay0), i.e., I at delay1 is considered the forecast for I at delay0, as represented by the equation [30]:(1)$$I_{t}=I_{t-1}$$

Persistence forecasting can be performed using simple methods such as taking the average or the most recent value of solar radiation or PV power output and using it as the forecast for the next time period. More sophisticated persistence methods may consider trends, seasonality, or cyclical patterns in the data. One advantage of persistence forecasting is that it requires minimal computational resources and can be easily implemented in real-time systems. It can also provide a useful baseline for comparing the performance of more complex forecasting methods. However, persistence forecasts can be inaccurate when the weather conditions change rapidly or when there are unexpected events that affect the PV system’s performance. Several studies have investigated the performance of persistence forecasting for PV systems. For example, it was found that persistence forecasts outperformed other simple methods such as autoregressive models for short-term PV power forecasting. However, persistence forecasts performed worse than more complex models such as artificial neural networks for longer-term forecasting. Overall, persistence forecasting can be a useful starting point for PV forecasting, especially for short-term predictions. However, it should be supplemented with more sophisticated methods for improved accuracy over longer time horizons or when weather conditions are changing rapidly [30].

Physical models are a widely used approach for forecasting PV power output. These models employ mathematical equations that represent the physical processes governing the conversion of solar radiation into electrical power in a PV system. Physical models can be highly accurate when the underlying physical processes are well understood and accurately modeled. However, they can be computationally intensive and may require detailed information on system parameters such as panel orientation, shading, and temperature. One common type of physical model used for PV power output prediction is the equivalent circuit model (ECM). The ECM uses electrical circuit equations to model the behavior of the PV system, taking into account factors such as the characteristics of the solar cells, the shading effects of nearby objects, and the thermal behavior of the system. Another physical model commonly used for PV power output prediction is the PV cell model. This model uses mathematical equations to describe the electrical and optical properties of the PV cell, taking into account factors such as the quantum efficiency, carrier recombination, and the effects of temperature and illumination [6,31]. Using PV panels as sensors in physical models can enhance their accuracy and adaptability to changing environmental conditions. By integrating real-time data from the PV panels, such as voltage, current, and temperature, the models can dynamically adjust to the system’s performance and environmental factors [29].

While statistical and machine learning models can be effective for PV power output forecasting, they may require large amounts of historical data and may not be as accurate in situations where the underlying physical processes are important. For example, statistical models and machine learning models may not capture the effects of shading or changes in the angle of incidence of sunlight on the PV panel [29].

The combination of physical and persistence models in PV power output forecasting has been shown to provide improved accuracy compared with either approach used alone. This hybrid approach takes advantage of the strengths of both physical and persistence models to produce more accurate forecasts. Physical models are highly accurate when they are well calibrated and validated and can capture the underlying physical processes that govern the conversion of solar radiation into electrical power in a PV system. However, physical models can be computationally intensive and may require detailed information on system parameters. Persistence models, on the other hand, are simple and computationally efficient and can provide a useful baseline for short-term forecasting when weather conditions are relatively stable. However, persistence models can be inaccurate when weather conditions change rapidly or when unexpected events occur. By combining physical and persistence models, it is possible to capture both the long-term trends and the short-term variations in the PV power output. For example, a physical model could be used to capture the long-term trends in PV power output based on weather conditions, while a persistence model could be used to capture the short-term variations based on the current or recent values of PV power output. Several studies have investigated the performance of hybrid physical–persistence models for PV power output forecasting. For example, it was found that a hybrid model based on a combination of a physical and a persistence model outperformed other models for short-term forecasting. Overall, the combination of physical and persistence models can provide improved accuracy for PV power output forecasting compared with either approach used alone [32], as it’s showed in Table 1.

## 4. PV-Panel-Based Forecasting

After our thorough examination of various forecasting methods and their respective advantages and disadvantages, our goal in this paper is to present an innovative and highly accurate forecasting approach. This method capitalizes on the use of identical PV panels as power sensors to monitor real-time changes in the power output. This method’s adaptability makes it especially suitable for solar-powered fencing, among other applications, demonstrating its practicality and the wide-ranging impact of our research in the renewable energy sector. By providing a forecasting solution that can be integrated with solar fencing systems, we address a critical need for reliable energy management in agricultural and security applications, underscoring the originality and utility of our approach. These panel sensors respond not only to changes in solar irradiance but also to variations in temperature, wind speed, aerosol distribution, and the accumulation of dust and dirt over time, all of which are critical factors affecting PV panel performance.

Consequently, any alteration in environmental conditions, such as cloud movement, that impact the sensors will similarly affect the main panels, allowing for a more accurate representation of the solar field’s response. In this paper, we outline the methodology for positioning the sensors, considering factors such as shading and panel orientation. Additionally, we detail the approach for calculating cloud movement and other environmental factors, as well as their influence on the overall power output of the solar field. This method can lead to improved nowcasting and forecasting models, which, in turn, can optimize system performance, reduce energy costs, and enhance grid stability.

In the proposed approach, a hybrid forecasting method is utilized, which combines elements of both persistence and physical forecast methods for predicting the impact of cloud cover on PV fields. The persistence method, a key component of this approach, is employed by observing the power drop in the sensor panels. This method assumes that the current trend will continue in the short term, providing essential information for immediate decision-making. On the other hand, the physical method incorporates a physical variable, specifically wind speed, into the forecasting process. By using basic geometry, the time it takes for the cloud to reach the main PV field can be calculated, thus enhancing the accuracy of the prediction. The combination of these two methods results in a simplified yet effective forecasting technique for ultra-short-term or nowcasting forecasts for PV fields. Although this hybrid method may not be as precise as more advanced statistical or machine learning techniques, it offers valuable insights for immediate decision-making and effective management of the PV system.

### Proposed Algorithm

Let us consider a field filled with rectangular solar panels arranged arbitrarily. The field is surrounded by sensors placed along the borders of some figure, with each sensor labeled by an index j. We introduce a coordinate system x and y, with the sensors having coordinates ($x_{j},\;y_{j}$). Each solar panel is numbered k, and all panels together form a single solar panel. It is assumed that each panel is small and can be covered by clouds or not. Let the clouds move at a constant speed *V* ($V_{x}$, $V_{y}$) with an arbitrary shape. At every moment of time $t_{i}$, the sensors indicate the presence or absence of a cloud. The objective is to determine the area of the panel covered by clouds, the length of time it is covered, and the power loss incurred, depending on the sensor readings. In this setting, the panels serve as an element for dividing the area of the solar panel.

In Figure 2, a sensor with index j is characterized by coordinates $x_{j},\;y_{j},$ and it has detected a shadow at the specific moment i. The point where the shadow is measured will intersect a panel element with index k at points A and B. Consequently, the coordinates of these intersection points are represented with the indexes i, j, and k.

We can define each panel using a system of inequalities.
(2)$$\left\{\begin{array}{c} X_{1k}\leq x\leq X_{2k}\\ Y_{1k}\leq y\leq Y_{2k} \end{array}\right.$$

The indexes with *k* denote the boundaries of the rectangular panel along the *X* and *Y* axes. Let us focus on one of the sensors, which measures the presence or absence of a shadow at a specific point with coordinates ($x_{ij},\;y_{ij}$). The index *i* represents the moment in time, while *j* corresponds to the number of sensors. The shadow emanates from the given point and travels along a straight line according to the following equation.
(3)$$y=y_{ij}+\frac{v_{y}}{v_{x}}\left(x-x_{ij}\right)$$

Assume that *V_x_* is not equal to zero. Let the points of intersection between line (2) and the lines representing the boundaries of the *k*-th panel be denoted as ($X_{1k},\;{y1}_{iijk}$), ($X_{2k},\;{y2}_{iijk}$), (${x1}_{iijk},\;Y_{1k}$), and (${x2}_{iijk},\;Y_{2k}$). Here, the index *ijk* denotes the coordinate of the intersection between panel *k* and the shadow detected by sensor *j* at time *i*. When *X*_1*k*_, *X*_2*k*_, *Y*_1*k*_, and *Y*_2*k*_ are known, we can calculate the remaining coordinates:(4)$$\left\{\begin{array}{c} y_{1iijk}=y_{ij}+\frac{v_{y}}{v_{x}}\left(X_{1k}-x_{ij}\right)\\ y_{2iijk}=y_{ij}+\frac{v_{y}}{v_{x}}\left(X_{1k}-x_{ij}\right)\\ x_{1iijk}=x_{ij}+\frac{v_{y}}{v_{x}}\left(Y_{1k}-y_{ij}\right)\\ x_{2iijk}=x_{ij}+\frac{v_{y}}{v_{x}}\left(Y_{2k}-y_{ij}\right) \end{array}\right.$$
assuming that *V_y_* is not equal to zero. Of these intersection points, only two points can lie on the boundary of the panel. If the intersection point happens to be in a corner, then both points coincide.

Moving on to the process of counting the shaded panels, let us define the matrix Hik, where each element represents the number of rays emitted from the boundary that crossed panel *k* at moment *i*. An intersection occurs if any two of the following conditions are met:(5)$$\left\{\begin{array}{c} Y_{1k}\leq y_{1iijk}\leq Y_{2k}\\ Y_{1k}\leq y_{2iijk}\leq Y_{2k}\\ X_{1k}\leq x_{1iijk}\leq X_{2k}\\ X_{1k}\leq x_{2iijk}\leq X_{2k} \end{array}\right.$$

Conditions (2)–(5) correspond to the placement of the points (*X*_1*k*_, *y*_1*iijk*_), (*X*_2*k*_, *y*_2*ijk*_), (*x*_1*ijk*_, *Y*_1*k*_), and (*x*_2*ijk*_, *Y*_2*k*_) on the edge of the panel. If any of these conditions are met, the panel can be considered covered. Note that the total area of shaded panels may exceed the sum of the panel areas for which Conditions (2)–(5) are satisfied at a given point in time because the coverage duration is also a factor.

We can estimate the duration of the intersection by recalling the coordinates of two of the four points (*X*_1*k*_, *y*_1*iijk*_), (*X*_2*k*_, *y*_2*ijk*_), (*x*_1*ijk*_, *Y*_1*k*_), and (*x*_2*ijk*_, *Y*_2*k*_) as (*x*_1_, *y*_1_) and (*x*_2_, *y*_2_). We can then calculate the moments of intersection.
(6)$${T1}_{ijk}=t_{i}+\sqrt{\frac{{{(x}_{1-}x_{ij})}^{2}-{{(y}_{1-}y_{ij})}^{2}}{{Vx}^{2}+{Vy}^{2}}}$$
(7)$${T2}_{ijk}=t_{i}+\sqrt{\frac{{{(x}_{2-}x_{ij})}^{2}-{{(y}_{2-}y_{ij})}^{2}}{{Vx}^{2}+{Vy}^{2}}}$$

We define ${T1}_{ijk}$ and ${T2}_{ijk}\;$ as tables of the times when the shadow emitted by sensor *j* at time *i* crosses the boundaries of panel *k*. Then, we compute *T_ijk_* as the maximum of *T*1*_ijk_* and *T*2*_ijk_*, which represents the duration for which the panel remains shaded. Within the matrix *Hik*, the term *h_ijk_* is defined as a binary indicator. Specifically, *h_ijk_* is set to 1 if the Conditions (2)–(5) are satisfied, indicating that the shadow detected by sensor *j* at time *i* is indeed covering panel *k*. Conversely, *h_ijk_* is set to 0 if the conditions are not met, signifying that panel *k* is not covered by a shadow at that time. Let *dS_k_* be the area of the *k*-th panel.

Using these definitions, we can write the condition for the coverage of panel *k* at time *i* = *I* as follows: if *tI* < *T_Ijk_*, then the shadow recorded by sensor *j* still covers panel *k* at time *I*. We check this condition for all *j* with a fixed *I* and *k*. If the condition is satisfied at least once, then we set *HIk* = 1; otherwise, we set it to 0. Here, *H_ik_* represents the coverage, and it can also be used to count the number of coverages for creating drawings.

The full coverage area is given by the equation
(8)$$S_{i}=\sum_{all\;k}H_{ik}dS_{k}$$
where $dS_{k}$ is the area of the *k*-th panel. The power loss at time *i* is the product of the solar constant *λ* and *S_i_*, where *λ* is determined by the weather conditions and geographic location.
(9)$$P_{i}{=\lambda S}_{i}$$

The total energy loss at a given point in time *i* = *I* is then computed as
(10)$$W_{i}=\sum_{i=0}^{i=I-1}P_{i}\left(t_{i+1}-t_{i}\right)$$

In this way, we can solve the problem of power losses for an arbitrary field of solar panels, which may be limited by sensors located at arbitrary positions, while also maintaining the constancy of the speed of the shadow.

## 5. Proposed Simulation Model

In this study, we present a comprehensive simulation model for a PV and lithium-ion (Li-ion)-battery-based energy storage system, incorporating DC/AC converters and a proportional–integral–derivative (PID) controller to regulate the charging and discharging processes. The model is developed using Simulink 2023, a powerful simulation and modeling software that enables the adjustment of irradiance levels to simulate the impact of cloud cover on the system. The proposed system is designed to feed an equivalent load of 22 kV while being connected to the national grid, ensuring a reliable and efficient energy management solution. The PV-battery system comprises four primary components: PV panels, a boost converter with an MPPT controller, a Li-ion battery, and power converters. The PV panels are responsible for converting solar energy into electrical energy, generating direct current (DC) electricity. This DC output is first optimized by the boost converter, which is controlled by an MPPT controller, before being transformed into AC by a DC/AC converter [33,34]. This makes the electricity suitable for consumption by the connected load or for feeding into the national grid [35,36]. The Li-ion battery serves as an energy storage system, storing any excess energy produced by the PV panels and supplying it to the load when needed. This is achieved through an AC/DC converter, which utilizes an average model to maintain computational simplicity while providing reliable performance. The integration of a PID controller in the system plays a crucial role in maintaining efficient energy management. The PID controller constantly adjusts the reference voltage of the converters, thereby controlling the charging and discharging processes of the Li-ion battery. This ensures that the battery operates within safe limits while maximizing its utilization and prolonging its lifespan. Furthermore, the PID controller contributes to the overall stability and reliability of the system, as it dynamically adapts to the varying energy demands and generation capabilities. In addition to the primary components, the system is connected to an equivalent load of 22 kV and the national grid, facilitating bi-directional power flow. This connection allows for seamless integration with the existing power infrastructure and provides a robust solution for addressing the intermittency and unpredictability of solar energy. By accommodating varying irradiance levels in the Simulink model, the system’s performance under different weather conditions can be analyzed, providing valuable insights for optimizing its efficiency and reliability. In the earlier section of this paper, we presented a hybrid algorithm for PV-power cast prediction, which combines the strengths of both physical and persistence methods while maintaining simplicity for quick calculations. To evaluate the effectiveness of this simple yet powerful algorithm, we plan to integrate it into the comprehensive simulation model shown in Figure 3 of the PV-battery system previously described. Our objective is to examine the algorithm’s potential to enhance the quality of electricity supplied to the load and the national grid by intelligently utilizing the energy storage system during weather-induced fluctuations in PV power generation. By anticipating decreases in the PV power output due to changing weather conditions, the algorithm can enable the PID controller to proactively manage the Li-ion battery’s charging and discharging processes. Consequently, the energy stored in the battery can be utilized to maintain a stable and reliable power supply when the PV power decreases. The integration of this simple hybrid algorithm aims to demonstrate its potential to optimize the performance of renewable energy systems by effectively leveraging energy storage in response to weather-related variations in power generation, ultimately contributing to the development of sustainable and resilient power infrastructure.

## 6. Verification

We evaluated the performance of the PV system under several distinct scenarios to comprehensively assess the effectiveness of the algorithm. The first scenario involves operating the system without utilizing energy storage at all. The second scenario incorporates energy storage; however, the algorithm is not employed, and the energy storage is activated only when clouds are already present in the field. In the third scenario, our algorithm serves as an early warning system, providing advance notice to the energy storage system before clouds reach the field, allowing it to proactively respond to impending fluctuations in PV power generation. The fourth scenario is similar to the third, but instead of giving early warning to the energy storage system, the grid operator is notified in advance, enabling them to assess and manage the power supply to the load accordingly. By comparing the outcomes of these scenarios, we aim to demonstrate the benefits and practical applicability of the proposed algorithm in optimizing renewable energy systems and ensuring a stable and reliable power supply.

Scenario 1: No PV-Battery System and with No Early Warning Algorithm for Energy Storage Activation and No Grid Operator Notification at a weak grid. In Figure 4, this scenario is graphically depicted to provide a straightforward view of how the system performs when it is directly exposed to the effects of passing clouds without any form of mitigation. This figure tracks the power output from the PV field (Ppv), the energy held in the battery (PBat), the power used by the load (PLoad), and the power sent to the grid (PGrid). The graph demonstrates the power output variability as clouds move across the PV field, underscoring the difficulty in maintaining a stable power supply to both the load and the grid when no storage or forecasting tools are in use. It highlights the necessity for precise forecasting to handle these fluctuations effectively.

In Figure 5, as depicted in scenario 1, a noteworthy aspect that warrants particular attention is the voltage drop that could potentially harm the load. This voltage drop poses a significant risk to the integrity and safety of the electrical system, including sensitive equipment and devices connected to the grid.

Scenario 2: PV System Without Energy Storage and with Early Warning Algorithm for Grid Operator Notification

In Figure 6, scenario 2 provides a comprehensive illustration of the dynamic relationship between cloud movement and PV power generation. This relationship is particularly significant for grid operators, as it necessitates timely responses to fluctuations in PV power output. This figure uses the blue line to represent the PV field power and is also a drawn map of a moving cloud passing over the PV field. This phenomenon can be divided into three different phases: *Initial Impact Phase*: As the cloud approaches and begins to cross the PV field, the power produced starts to decrease. This decrease is contingent on the cloud’s velocity and density, two variables that can significantly influence the rate of power reduction. *Cloud Crossing Phase*: During this phase, the cloud continues to traverse the PV field. The power decrease persists, reflecting the cloud’s ongoing obstruction of sunlight. This phase emphasizes the importance of accurate cloud forecasting, as it allows the grid operator to anticipate and respond to these changes in real time. *Cloud Exit Phase*: This final phase occurs when the cloud crosses out of the field and passes the sensors. The power begins to stabilize, returning to its pre-cloud levels.

The red line in the figure illustrates the grid power changes in response to the PV power fluctuations. This line is indicative of the grid’s adaptability and responsiveness, key attributes in maintaining stability in the face of variable renewable energy sources.

The yellow line, representing the power over the load, and the brown line, indicating the absence of active energy storage, further contribute to the overall understanding of the system’s behavior. The absence of active energy storage, as shown by the brown line, underscores the grid’s reliance on real-time adjustments to compensate for changes in power generation.

The figure demonstrates that the grid compensates for the lack of power by pushing slightly more power forward to the loads. This compensation is not merely a reactive measure; it is a calculated response that ensures that the grid values and the grid inertia remain within the allowed range. This highlights the importance of integrated planning and real-time control in modern PVs and grids.

Figure 7, depicting scenario 2, provides an insightful examination of the voltage response to the extra power that the grid is building due to forecasting. This figure emphasizes a critical aspect of modern grid management, particularly in the context of renewable energy integration. The primary significance of this figure lies in the demonstration that the voltage remains within the allowed boundaries. This stability is not merely a technical detail; it is a fundamental requirement for the safe and efficient operation of the electrical grid. Any deviation from these boundaries could lead to system instability or even failure, making the control of voltage an essential task for grid operators. The figure also reveals an interesting dynamic at the end of the cloud’s movement, which commences at the conclusion of its crossing over the PV fields. Here, the grid power is directed according to the forecasting to the other side of the PV field. This maneuver is indicative of the grid’s adaptability and the vital role that accurate forecasting plays in optimizing the energy distribution. It is essential to recognize the broader context in which this scenario occurs. Most PV fields lack a storage system, a reality that presents both challenges and opportunities. Implementing large-scale storage is often prohibitively expensive, a factor that can limit the flexibility and resilience of the grid. In this context, forecasting emerges as a valuable solution, offering a cost-effective means to mitigate grid fluctuations.

Scenario 3: PV-Battery System Without an Early Warning Algorithm for Energy Storage Activation and No Grid Operator Notification

Figure 8 illustrates scenario 3, which focuses on a PV field equipped with an energy storage system. This scenario offers a nuanced view of how the storage system interacts with the grid, particularly during significant power fluctuations. The figure demonstrates the storage system’s response to a substantial drop in power. As the PV field’s power output declines, the storage system begins to deliver power according to a predetermined decision algorithm. This response is not instantaneous; the figure shows a corresponding drop in both grid power and load power until the entire system can respond. This delay underscores the complexity of integrating storage into the grid and the importance of carefully coordinated control mechanisms. A critical consideration in this scenario is the size of the storage system. The figure shows that storage capacity is inherently limited, and designing a system to handle the worst-case scenario can be prohibitively expensive. This constraint necessitates a balanced approach, one that considers both capacity and cost. An additional point of interest in Figure 6 is the large variation at the cutoff of the storage system. Since the scenario focuses on handling forecasting only, this variation presents a challenge. However, it is worth noting that this problem can be addressed through a dedicated control system for the storage’s end operation. Such a system would provide more precise control over the storage’s discharge, enhancing both efficiency and reliability.

Scenario 4: PV-Battery System With Early Warning Algorithm for Energy Storage Activation and No Grid Operator Notification

Figure 9 illustrates scenario 4, which focuses on a PV field that is equipped with an energy storage system. This scenario provides a detailed examination of how the storage system operates in conjunction with forecasting information, particularly in response to impending cloud cover. In this scenario, the storage system begins delivering power based on the information provided by the forecasting system, acting in the short term before the clouds start to shed. This proactive response is a key feature of the scenario, and it has a notable impact on the system’s behavior. Unlike other scenarios where a drop in power might be observed, here, the load remains almost unaffected. This stability is a testament to the effectiveness of the combined use of storage and forecasting, enabling a more resilient and responsive energy system. However, it is essential to recognize the challenges and limitations associated with energy storage. Storage systems are not only expensive to purchase but also come with ongoing maintenance costs. Their cycles are limited, and they are often used for specific purposes, such as storing energy during the day and selling it at a higher price at night. These factors add complexity to the decision-making process around storage and must be carefully considered in the design and operation of a PV system with storage.

Scenario 5: PV-Battery System With Reactive Energy Storage Activation during Successive Cloud Transitions.

In scenario 5, depicted in Figure 10, we examine the system’s reaction to consecutive cloud cover transitions affecting the PV field. The blue line, charting the PV output (Ppv), dips with each cloud’s passage, showing the system’s quick response to shifting shading. The red line tracks the battery power (PBat), demonstrating how the energy storage compensates for PV output fluctuations. Being activated with each cloud, the energy storage system helps to steady the photovoltaic power supply. The inset zooms in on the system’s forecasting ability, which accurately predicts cloud coverage and prompts the energy storage response. This functionality is key to the system’s proactive control and ability to maintain power supply stability amidst environmental changes.

### Voltage Response across Different PV System Scenarios

Figure 11 displays the consolidated voltage response of the photovoltaic (PV) system under five different operational scenarios, each differentiated by color. This composite graph allows for a straightforward comparison of how the system’s voltage stability is influenced by cloud cover events and the implementation of energy storage and forecasting algorithms. Scenario 1, depicted by the blue line, shows the system’s voltage without any form of energy storage or forecasting. This scenario indicates the inherent instability in the absence of mitigation strategies. Scenario 2, represented by the red line, introduces energy storage that activates upon cloud detection. Here, the voltage response improves slightly, suggesting a reactive benefit to voltage stability. Scenario 3, depicted by the green line, features energy storage without the use of forecasting. This scenario’s voltage response indicates a different pattern of stability when the system can store and discharge energy but lacks predictive capabilities. Scenario 4, shown by the black line, combines energy storage with a forecasting algorithm, offering a more proactive approach to voltage stabilization. Scenario 5, marked by the cyan line, demonstrates the system’s behavior with reactive energy storage in response to successive cloud transitions. This scenario tests the system’s robustness in a dynamic environment, simulating the effect of rapidly changing cloud cover. In all, the figure underlines the importance of both predictive forecasting and energy storage in maintaining voltage stability against the variability of cloud cover, which is critical for ensuring the reliability and safety of the grid-connected PV system.

Verification and Statistical Analysis. To evaluate the impact of the proposed forecasting method on voltage stability, we conducted a simulation study that compares two scenarios: (1) a PV system without forecasting and (2) a PV system with forecasting. Five independent simulations were run for each scenario, and the average power output and standard deviation were recorded for each simulation. An independent samples *t*-test was performed to assess the difference in average power output between the two scenarios. The calculated *p*-value was 0.001174, indicating a statistically significant difference (*p* < 0.05). This suggests that the forecasting method has a significant impact on the average power output of the PV system. Furthermore, a power analysis was conducted to determine the adequacy of the sample size. With a desired power of 0.8 and a significance level of 0.05, the analysis indicated that a sample size of 5 per group was sufficient to detect a statistically significant difference, given the effect size calculated from the data. These findings provide evidence that the proposed forecasting method can significantly improve the average power output of a PV system, contributing to enhanced voltage stability and overall system performance.

## 7. Conclusions

In conclusion, this scientific article presents a novel, simple hybrid algorithm for PV power cast prediction, which combines the strengths of both physical and persistence methods while enabling quick calculations. The algorithm was successfully integrated into a comprehensive simulation model of a PV-battery system, which included Li-ion battery energy storage, DC/AC converters, and a PID controller. The objective of this integration was to evaluate the algorithm’s potential to enhance the quality of electricity supplied to the load and the national grid by intelligently utilizing the energy storage system during weather-induced fluctuations in PV power generation.

The results demonstrate that the algorithm effectively anticipated decreases in PV power output due to changing weather conditions, enabling the PID controller to proactively manage the Li-ion battery’s charging and discharging processes. Consequently, the energy stored in the battery was utilized to maintain a stable and reliable power supply when the PV power decreased. This successful implementation and evaluation of the simple hybrid algorithm showcases its potential to optimize the performance of renewable energy systems by effectively leveraging energy storage in response to weather-related variations in power generation. Furthermore, the statistical analysis conducted in the verification section, which yielded a *p*-value of 0.001174, confirms the significant impact of the proposed forecasting method on the average power output and voltage stability of the PV system, demonstrating its effectiveness quantitatively. The evaluation of the different scenarios demonstrated that without energy storage (Scenario 1), the system faced significant instability, emphasizing the need for effective forecasting. Incorporation of energy storage without the algorithm (Scenario 2) provided some relief, but the system still relied heavily on real-time grid adjustments. The use of the forecasting algorithm as an early warning system (Scenario 3 and 4) showed marked improvements in managing power supply and grid stability, highlighting the algorithm’s efficacy. Finally, Scenario 5, involving reactive energy storage during successive cloud transitions, demonstrated the system’s robustness and the critical role of forecasting in dynamic weather conditions.

This solution is far more important when there is no storage in the PV field as it is of low cost and provides a good response to short-term fluctuations; in this way, it helps the grid to be maintained at nominal values.

Furthermore, by highlighting the potential integration of our forecasting method with solar-powered fencing, we underline a significant advancement in the application of solar energy technologies. This aspect of our research not only broadens the scope of PV power forecasting but also showcases the practical implications of our work in enhancing the sustainability and efficiency of solar energy systems.

Overall, this study contributes to the development of sustainable and resilient power infrastructure and offers valuable insights for researchers and practitioners in the field of renewable energy and smart-grid technologies.

## Figures and Tables

**Figure 1 sensors-24-02866-f001:**
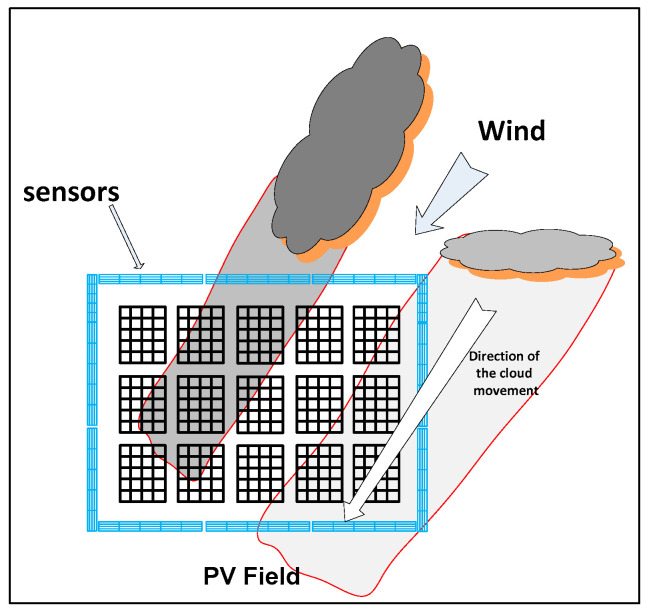
System under consideration.

**Figure 2 sensors-24-02866-f002:**
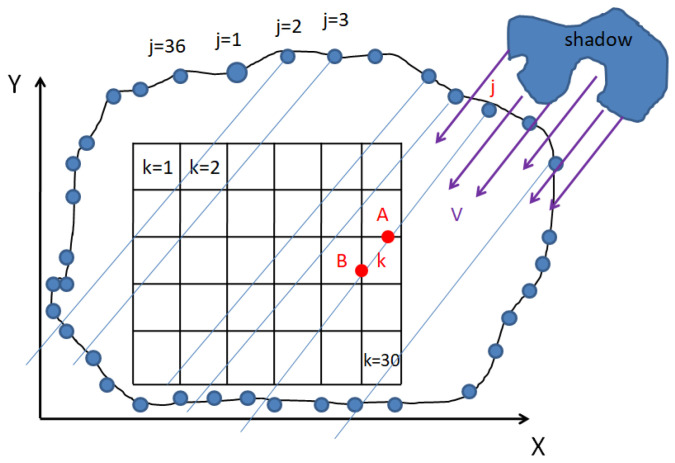
Details of the system components on a system of axes for the purpose of mathematical calculations.

**Figure 3 sensors-24-02866-f003:**
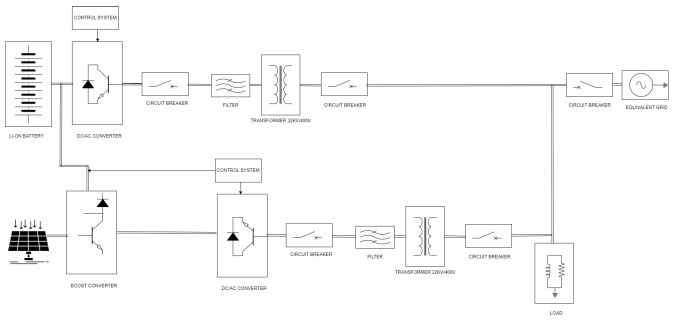
Schematic of the comprehensive simulation model of a PV and battery storage system.

**Figure 4 sensors-24-02866-f004:**
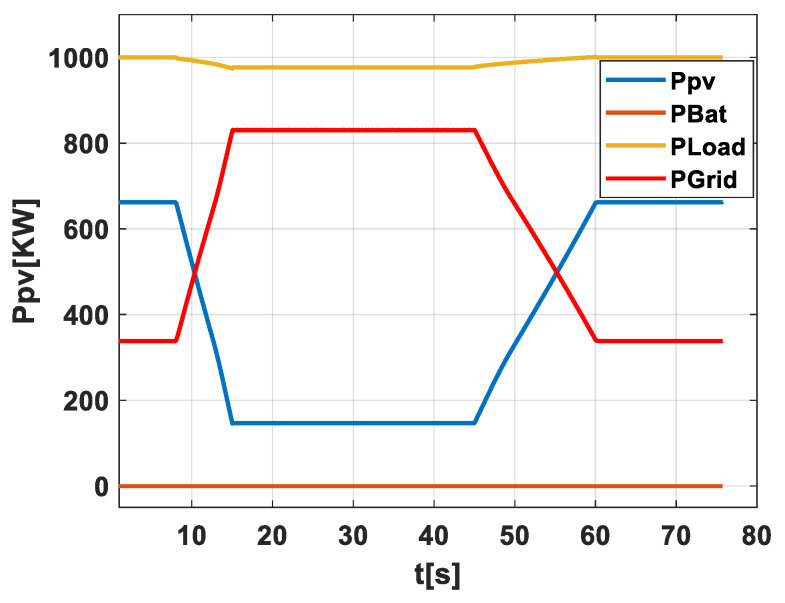
Illustration of scenario 1.

**Figure 5 sensors-24-02866-f005:**
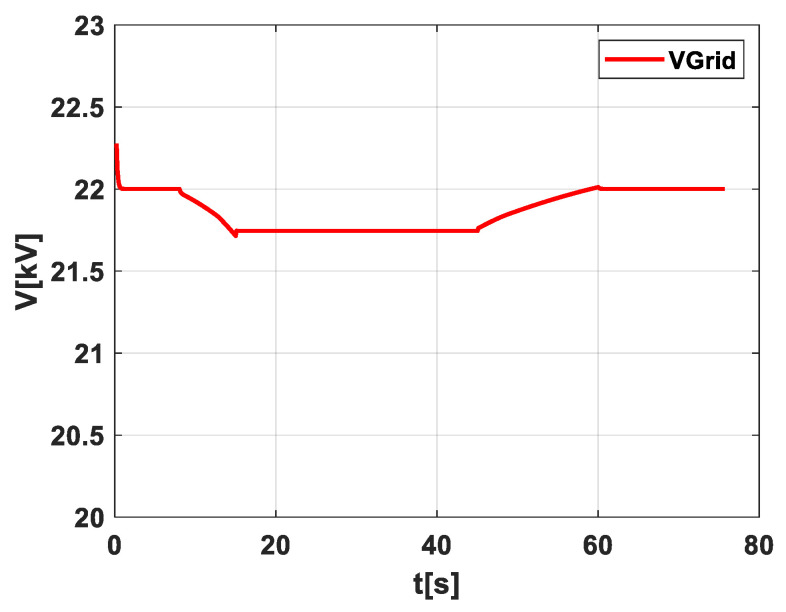
Illustration of the voltage behavior in scenario 1.

**Figure 6 sensors-24-02866-f006:**
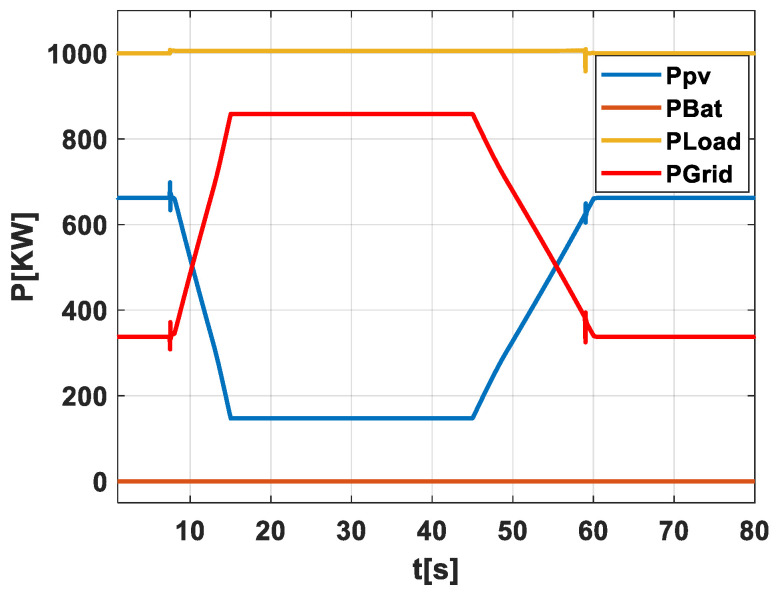
Illustrates scenario 2.

**Figure 7 sensors-24-02866-f007:**
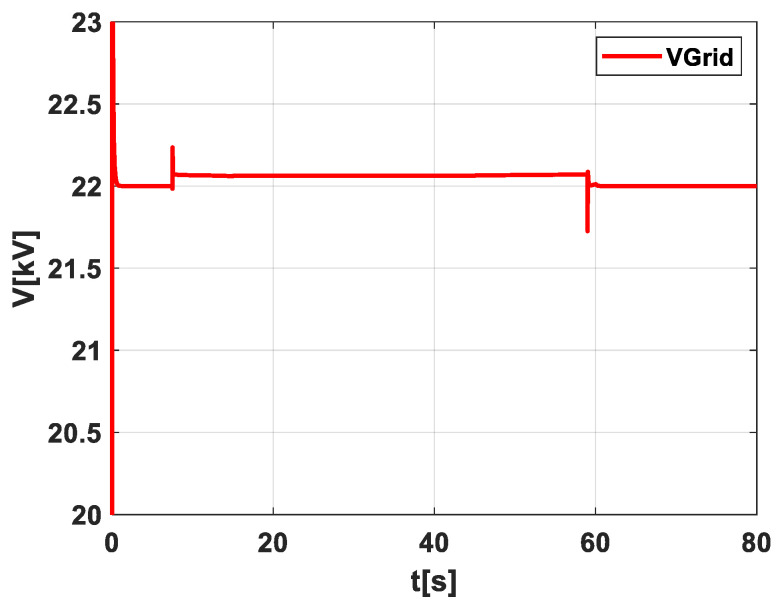
Illustration of the voltage behavior in scenario 2.

**Figure 8 sensors-24-02866-f008:**
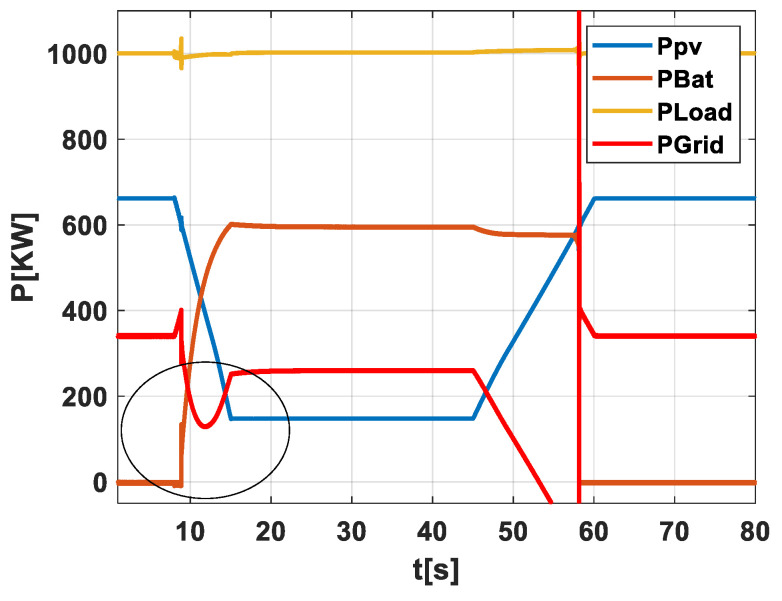
Illustration of scenario 3.

**Figure 9 sensors-24-02866-f009:**
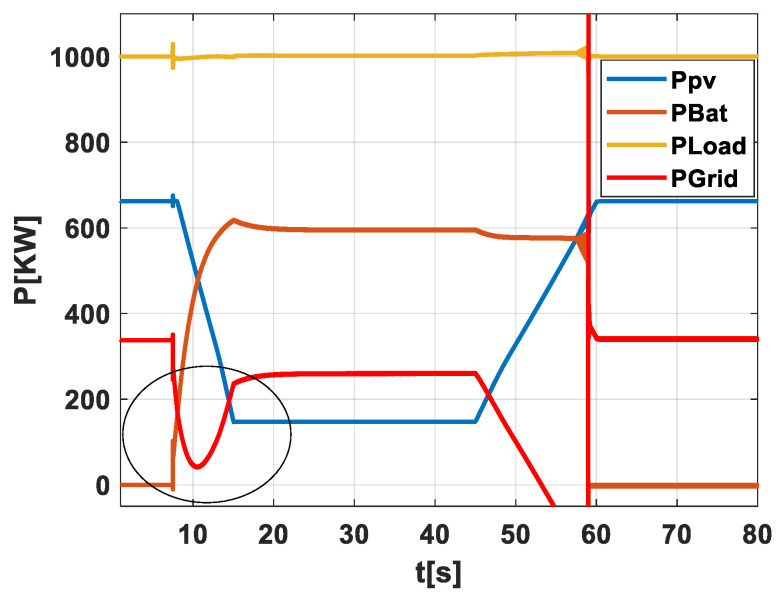
Illustration of scenario 4.

**Figure 10 sensors-24-02866-f010:**
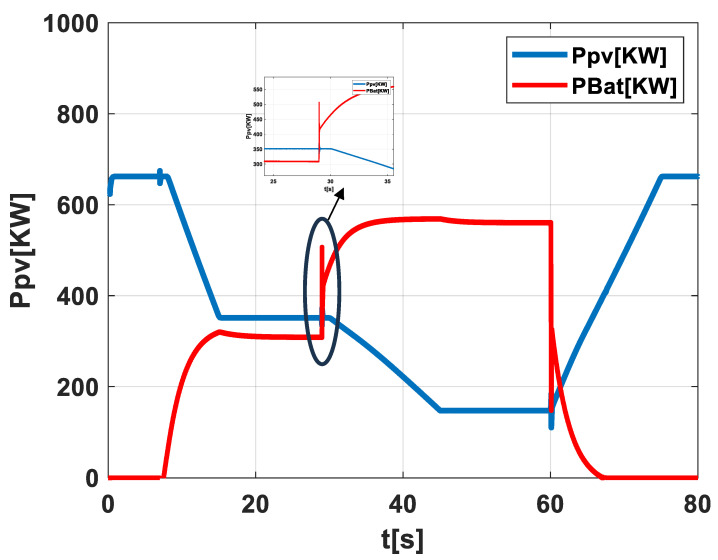
Illustration of scenario 5.

**Figure 11 sensors-24-02866-f011:**
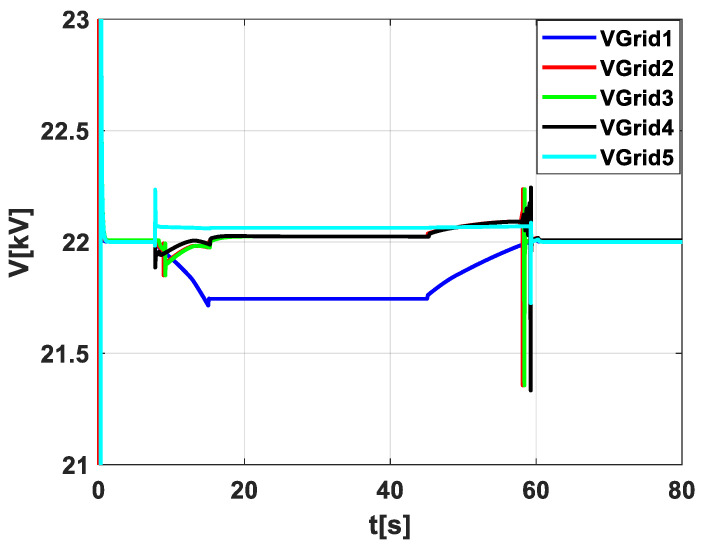
Comparative voltage profiles in PV system scenarios.

**Table 1 sensors-24-02866-t001:** Comparison of forecasting models for PV power output prediction.

Forecast Model Type	Advantages	Disadvantages
Persistence Models	Simple and computationally efficient; require minimal computational resources; can be easily implemented in real-time systems; provide a useful baseline for comparing more complex forecasting methods.	Can be inaccurate when weather conditions change rapidly or during unexpected events; generally perform worse than more complex models for longer-term forecasting.
Physical Models	Can be highly accurate when underlying physical processes are well understood and modeled; use real-time data from PV panels to enhance accuracy and adaptability.	Computationally intensive; require detailed system parameter information; limited by spatial and temporal resolutions.
Statistical Models	Use historical data and mathematical techniques to predict PV power output based on input and output variable correlations; potentially more straightforward than machine learning models.	May not be as accurate in situations where understanding the underlying physical processes is important; require large amounts of historical data.
Machine Learning Models	Adapt predictions over time by learning from large datasets; potentially provide more accurate forecasts as they learn from new data.	Require significant amounts of historical data; may not capture effects of shading or changes in sunlight incidence angle accurately without an understanding of the physical processes.

## Data Availability

Data are contained within the article.

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
