# Peer review of "A Combined Persistence and Physical Approach for Ultra-Short-Term Photovoltaic Power Forecasting Using Distributed Sensors"

_sensors, 2024, doi:10.3390/s24092866_

Round 1
Reviewer 1 Report
Comments and Suggestions for Authors
After thoroughly examining your paper, I have come across several substantial problems within its content:
1-It is important to place references at the end of the sentence.
2- Introduction can be better motivated. What is the really contribution of this work compared to the existing studies? What are the gaps in the existing literature.
3- Emphasis on big general ideas including originality and clearer contribution in relation to the literature : Making your research appealing to a broad audience is an important goal of this journal.
4-The literature review should be updated by including more recent studies on this topic, specifically studies from hybrid system.
5-It's more effective to present the old methods in a categorized table with their advantages and disadvantages instead of presenting them all together in the introduction.
6- Please cite all the equations that were adopted from a reference.
7- The authors do not provide detailed explanations on the calculation of PI control parameters and the in their work.
8- Please analyze the robustness of the hybrid system in response to the parameter uncertainties.
9-Enhancing the Results and Discussion section and discussing findings in relation to previous studies, literature, theory, and actual evidence is necessary.
Comments on the Quality of English LanguageThe quality of English language in the paper requires improvement. There are several grammatical errors, awkward sentence constructions, and inconsistencies in vocabulary usage throughout the text.
Reviewer 2 Report
Comments and Suggestions for Authors
The authors have worked on the Article "A Combined Persistence and Physical Approach for Ultra-Short-Term Photovoltaic Power Forecasting Using Distributed
Sensors". The work seems interesting; however, a few modifications and details are needed to be added before considering it for publication.
Abstract: The section needs to be written more quantitatively.
Introduction: Please elaborate on the section and add a few recent works in detail to demonstrate the reason for carrying out the research. Also, mention some disadvantages of previous research.
Here are a few suggestions:
What is the novelty of the work? Needs to be clearly mentioned in the article.
Result:
Author are requested to add a comparative data with the previously research to show the novelty of work.
Conclusion: More quantitative with future scope and area for application.
Reviewer 3 Report
Comments and Suggestions for Authors
In the manuscript, the author represents: “A Combined Persistence and Physical Approach for Ultra-Short-Term Photovoltaic Power Forecasting Using Distributed Sensors”. Moreover, the author needs to edit and answer some issues:
1. Fig. 1 and 2 need to combine into one.
2. The author needs to show the P results with the condition of the weather.
3. The author needs to double-check the caption of Fig. 4. The resolution of Fig. 4 is low.
4. The author needs to show figures of “voltage in scenarios: 3 and 4” and combine them (1, 2, 3, and 4) into one.
5. Why is time (t) from 0 to 80s?
6. The author needs to show the results of equations: from (1) to (8).
Reviewer 4 Report
Comments and Suggestions for Authors
This is a very well-written Manuscript, with minor editing errors. The authors described the research problem well and performed different scenario calculations that were validated on a simulated object. The calculations also seem to have been carried out correctly. It seems to me that the work can be published after small corrections.
The Introduction Chapter is too short.
I would like to remind you that the Introduction Chapter should present the state-of-the-art.
Figure 1 could be bigger and it could consist of more details.
Figure 1 is not mentioned in the text.
Lines 107-115 look to me like the repetition of lines 65-71.
Points 3.1, 3.2, 3.3., and 3.4 are again a repetition of lines 107-115, more detailed. Can't this be discussed straight away in lines 65-71?
Figure 2 is not mentioned in the text,
Line 369 “hijk” ? Could you explain this?
Line 386 – “dc/ac” it should be DC/AC.
Line 384, point 5 - does the model take into account partial shading of the PV panel field? Is this field divided into separate strings and integrated with inverters?
Figure 4 is not mentioned in the text.
Figure 4 is too small.
Figure 5 is not mentioned in the text.
Reviewer 5 Report
Comments and Suggestions for Authors
Missing experimental results. Put some preliminary experimental results and compare them with simulation results.
Improve the quality of the figure 4.
Put more actual paper from MDPI Journals in the references list.
Round 2
Reviewer 1 Report
Comments and Suggestions for Authors
I would like to express my sincere gratitude for the effort made by the authors in the correction: All the comments have been corrected.
Comments on the Quality of English LanguageI would like to express my sincere gratitude for the effort made by the authors in the correction: All the comments have been corrected.
Author Response
Thank You for the review
Reviewer 3 Report
Comments and Suggestions for Authors
1. The author needs to combine 2 figures (Fig.1 and 2) into 1 figure, please.
2. I'm sorry, I can not find new P result in manuscript.
3. The author needs to show figures of “voltage in scenarios: 3, 4 and 5”, please.
4. What happens if the time is increased or decrease (t<80s or t>80s)?
5. The author needs to double check and explain equation 1.
Reviewer 5 Report
Comments and Suggestions for Authors
Missing experimental validation.
